# Determination of variable region sequences from hybridoma immunoglobulins that target *Mycobacterium tuberculosis* virulence factors

**Hui-Chen Chang Foreman**[¤a‡*], **Andrew Frank**[¤b], **Timothy T. Stedman**[‡*]

BEI Resources, ATCC., Manassas, Virginia, United States of America

¤a Current address: Praedicare Inc., Farmers Branch, Texas, United States of America
¤b Current address: The Informatics Application Group, Inc. (TIAG), Reston, Virginia, United States of America
‡ Co-senior authors.
* jhcforeman@gmail.com (HCCF); tstedman@atcc.org (TTS)

**Data Availability Statement:** All relevant data are within the paper and its Supporting Information files.

## Abstract

*Mycobacterium tuberculosis* (Mtb) infects one-quarter of the world's population. Mtb and HIV coinfections enhance the comorbidity of tuberculosis (TB) and AIDS, accounting for one-third of all AIDS-associated mortalities. Humoral antibody to Mtb correlates with TB susceptibility, and engineering of Mtb antibodies may lead to new diagnostics and therapeutics. The characterization and validation of functional immunoglobulin (Ig) variable chain (IgV) sequences provide a necessary first step towards developing therapeutic antibodies against pathogens. The virulence-associated Mtb antigens SodA (Superoxide Dismutase), KatG (Catalase), PhoS1/PstS1 (regulatory factor), and GroES (heat shock protein) are potential therapeutic targets but lacked IgV sequence characterization. Putative IgV sequences were identified from the mRNA of hybridomas targeting these antigens and isotype-switched into a common immunoglobulin fragment crystallizable region (Fc region) backbone, subclass IgG2aκ. Antibodies were validated by demonstrating recombinant Ig assembly and secretion, followed by the determination of antigen-binding specificity using ELISA and immunoblot assay.

## Introduction

*Mycobacterium tuberculosis* (Mtb) is the causative agent of the transmissible respiratory disease Tuberculosis (TB) and is considered one of the most insidious and intractable pathogens in history. Mtb can synergize destructively with other infective agents, including HIV [1, 2]. Mtb/HIV coinfections account for one-third of all AIDS-associated mortalities [3]. The World Health Organization and the National Institutes of Health consider TB a critical health priority [4, 5]. As with other bacterial pathogens under antibiotic pressure, the incidence of multi-drug and extensively drug-resistant Mtb is on the rise [6].

Diversified approaches are necessary for the detection and control of TB. Recent evidence points to humoral antibody (Ab) correlating well with TB susceptibility, suggesting recombinant Ab against Mtb are viable therapeutic strategies [7–14]. The complexity of Mtb infection

**Funding:** This work was funded under contract HHSN272201600013C by the National Institute of Allergy and Infectious Diseases, National Institutes of Health, Department of Health and Human Services. The views expressed in this publication neither imply review nor endorsement by HHS. The funders had no role in study design, data collection and analysis, decision to publish, or preparation of the manuscript.

**Competing interests:** The authors have declared that no competing interests exist.

and progression, however, hampers development of universal TB diagnostics or therapeutics. The pathogenesis of TB can vary from an active, open form of pulmonary TB to an asymptomatic, closed-form of latent TB [15, 16]. While host-pathogen interactions are diverse throughout infection and pathogenesis, Mtb remains replication-competent [17]. Therefore, a reasonable strategy to develop diagnostic and therapeutic antibodies would target the virulence-associated factors of Mtb.

Cellular components critical for development and pathogenesis help define virulence-associated factors. Mtb transmission occurs by aerosol and establishes infection in the host lung after macrophage infiltration [16, 18]. Mtb circumvents macrophage degradation from lysosomal hydrolases and reactive oxygen and nitrogen species by inhibiting phagosome-lysosome fusion or maturation [19–21] and escapes the acidic environment of the phagolysosome [19, 22]. Instead of clearing the mycobacterium bacilli, macrophages become the major Mtb reservoir [23, 24]. Thus, one of the hallmarks of infection with Mtb is its persistence within arrested, immature macrophage phagosomes [18, 25]. Virulence-associated factors are likely associated with Mtb infiltration or persistence.

The number of antibody sequences known for Mtb antigens, virulence-associated or not, is limited [26–28]. Two major groups of virulence-associated Mtb factors have been identified through gene inactivation in a validated TB model correlating to a measurable loss in Mtb macrophage residence or Mtb fitness [16, 18, 25, 29–33]. KatG, a catalase-peroxidase enzyme [16, 29], and SodA, a member of superoxide dismutase complex [16, 31], are key elements for bacterial redox homeostasis to counteract host macrophage reactive oxygen and nitrogen species [30]. PhoS1/PstS1 is a phosphate-binding factor within the ABC phosphate transporter system, important for nutrient import and drug export [32, 34]. PstS1 also acts as a mycobacterial cell adhesin, promoting macrophage phagocytosis [34, 35]. The heat shock proteins GroES and GroEL are conserved chaperone proteins [16, 36, 37], necessary for proper protein folding. Both PhoS1 and GroES stimulate host B cell and T cell immunity as immunodominant antigens and are considered TB vaccine candidates [38–41]. SodA and KatG antibodies could develop into diagnostic or therapeutic tools (e.g., inhibiting Mtb survival). All four targets offer some promise for further development.

The first step in such development is the identification of the sequence encoding the Immunoglobulin (Ig) variable region (IgV or Fab region). Structurally, the IgV region in antibodies is composed of portions of the heavy ($IgV_H$) and light ($IgV_L$) chains [42, 43]. The antigen-binding region within the IgV is constructed via three complementarity-determining regions (CDRs), flanked by four framework regions (FR 1–4) [42, 44]. CDRs are in the loop region of IgV, directly participating in antigen-recognition specificity and affinity, and are the key determinants of immune antigen-recognition diversity [45]. The remainder of the Ig, the constant or Fc region, determines the effector function within an Ab. While PCR can amplify Ig sequences for functional antibodies in hybridomas [46–49], a plethora of highly homologous but aberrant Ig transcripts confound interpretation [49–53].

In this study, we deciphered and validated sequences corresponding to the $IgV_H$ and $IgV_L$ chains of immunoglobulin expressed in hybridomas to Mtb-SodA, Mtb-KatG, Mtb-GroES, and Mtb-PstS1. Using 5' Rapid Amplification of cDNA End (RACE) PCR, we amplified the CDR1-3 and FR 1–4 (IgV) regions. We sequenced the IgV amplicons through traditional cloning and Sanger sequencing and compared this approach with the deep sequencing ability of Next Generation Sequencing (NGS) to identify all potential IgV sequences. Aberrant IgV chains were systematically identified and eliminated using bioinformatics methods. Retention of paratope-determining sequences were confirmed by isotype-switching the putative IgV in an Fc-recombinant backbone. We validated each recombinant IgV for comparable antigen-binding activity to the target antigens as that from antibodies secreted by the parental hybridomas. We have

deciphered sequences with this validation method corresponding to the $IgV_H$ and $IgV_L$ chains of immunoglobulin to Mtb-SodA, Mtb-KatG Mtb-GroES, and Mtb-PstS.

## Results

### Sequencing of the murine $IgV_H$ and $IgV_L$ regions of hybridoma transcripts

5' RACE-PCR was employed to isolate IgV sequences expressed in hybridoma mRNAs. Since an isotype-specific antisense primer is required for the PCR assay, identification of hybridoma isotypes was integral prior to IgV sequencing. Isotypes of four hybridoma clones were determined by an anti-mouse, isotype-specific, lateral flow assay: IgG1κ for PhoS1/PstS1[NRC-2410], IgG1κ for SodA[NRC-13810], IgMκ for KatG[NRC-49680], and IgG2aκ for GroES[NRC-2894] (**S1A Fig**). The isotype class enabled the design of the 3' isotype-specific primer (3' ISP) (**Tables 1 and S1**).

High-quality RNA preparations were produced to ensure the isolation of contiguous IgV domains. Extracted RNA was evaluated for purity by absorbance at A260/A280 (>1.9), the integrity of 28S/18S rRNA (>2), and measurement of RNase contamination after 37˚C treatment (retention of 28S band) (**S1B–S1D Fig**). RNAs satisfying all three criteria were used for $1^{st}$ strand cDNA synthesis and subsequent IgV RACE-PCR amplification.

The diversity of antigen recognition encoded by IgV is one of the hallmarks of immunity and requires considerable sequence variability. IgV sequences were generated by 5' RACE-PCR (Takara Inc., Mountain View, CA), using a 3'-antisense primer specific to conserved isotype-specific constant regions. The isotypes of the hybridoma clones included 4 different subclasses, IgG1, IgG2a, and IgM for Ig heavy chains and Igκ for Ig light chains. The respective 3' isotype-specific primer sequences are highly conserved (>97% pairwise identity) (**S1 Table**). Sequence alignment was performed for the corresponding CH1-constant region of heavy chains and the CH-constant region of light chains as downloaded from the IMGT/LIGM-DB database [54]. The IMGT/LIGM-DB database collectively contains immunoglobulin sequences deposited to INSDC by participating nucleotide databases [55]. The anticipated RACE-PCR-$IgV_H$ (~650 bp) and -$IgV_L$ (~550 bp) amplicons encompass an Ig leader sequence and full-length CDR1-3/FR1-4 (IgV) sequences (**Fig 1A**). The immunoglobulin heavy and light chain fragments of the 5' RACE-PCR products were detected and isolated using agarose gel electrophoresis (**Fig 1B**). Though the amplicons of $IgV_H$ and $IgV_L$ appeared as a homogenous, single band on an agarose gel, direct Sanger sequencing of the eluted DNA fragments indicated a mix of Ig templates (**S7 Fig**), confirming the presence in the hybridomas of sequences encoding aberrant Ig chains.

### Direct TOPO cloning and Sanger sequencing of murine $IgV_H$ and $IgV_L$ regions

We examined the Ig-transcript profile of IgVs by cloning and sequence analysis of the RACE-PCR amplicons. The in-gel IgV DNA amplicons were extracted and purified for

**Table 1. List of the 3' isotype-specific primer (ISP) sequences in this study.**

| Name of 3'ISP*s | Isotype Targets (*Mus musculus*) | 5' to 3' nucleic acid sequence |
|---|---|---|
| mIgG1 | IgG1 | CCGCTGGACAGGGATCCAGAGTTCCAGG |
| mIgG2a | IgG2a | CCACTGGACAGGGATCCAGAGTTCCAGG |
| mIgκ | Igκ | GGATACAGTTGGTGCAGCATCAGCCCG |
| mIgM | IgM | CAGGTGAAGGAAATGGTGCTGGGCAGG |

*: 3'-isotype-specific primer.

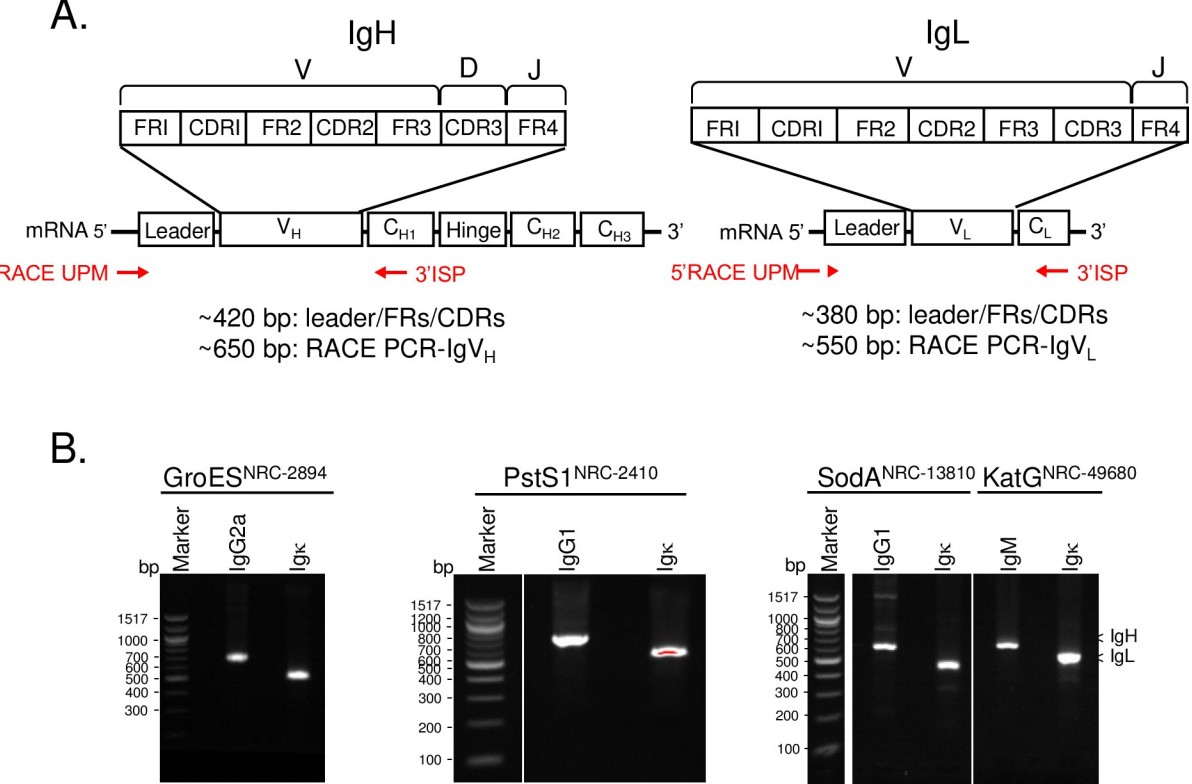

**Fig 1. 5' RACE-PCR immunoglobulin heavy (IgV$_H$) and light (IgV$_L$) chain products for each of the 4 hybridoma cell lines. A.** Illustration of the 5' RACE-PCR amplicons of IgV$_H$ and IgV$_L$ used in this study. The 5' RACE Universal Primer (5' RACE UPM) and antisense, 3' isotype-specific primer (3' ISP) were marked. **B**. Agarose gel electrophoresis of the 5' RACE-PCR amplicons of IgV$_H$ and IgV$_L$ for the 4 hybridoma cell lines. The amplicons were separated by 1.5% low melting agarose electrophoresis with ethidium bromide and were excised, extracted, and column purified. The corresponding amplicons of IgV$_H$ or IgV$_L$ were used either for generating an NGS library for MiSeq Illumina sequencing or for cloning into a TOPO-TA vector, followed by Sanger sequencing.

making an IgV library for Illumina next generation sequencing (NGS) or for cloning into a TOPO-TA vector (Invitrogen Inc.) for Sanger sequencing. Amplicons were cloned, and multiple independent clones (7 to 10) from each hybridoma Ig-amplicon were sequenced using the Sanger method. The resulting Ig sequences were analyzed through multiple sequence alignment to determine homology, translated to assess open reading frames, and aligned to *M. musculus* VDJ genes with MiGMAP [56]. In addition to "productive" Ig transcripts (i.e., containing open reading frames), multiple "unproductive" or "aberrant" transcripts (ORFs disrupted with stop codons) were also found (**S2 Fig**).

## Next generation sequencing and *de novo* transcript assembly of hybridoma IgV$_H$ and IgV$_L$ sequences

To enable a robust IgV sequencing approach, we compared cloning and Sanger sequencing results with deep sequencing directly from amplified Ig transcripts. MiSeq Illumina NGS coverage and high-fidelity sequences are limited to a continuous read length less than 300 bp, shorter than the IgV PCR amplicons (~500–600 bp), and assembly of reads into the complete IgV amplicon sequence is required. Selecting NGS contig assembly parameters, thresholds, or filtering schemes was challenging due to the highly homologous nature of aberrant transcripts but benefitted from the contiguous amplicon sequence information generated from Sanger

sequencing. Sanger reference sequences guided Ig-NGS sequencing methodology with proper filters, thresholds, and metrics and provided cross-validation of the assembled transcript candidates.

Eight libraries from 5' RACE-PCR IgV amplicons of the hybridoma heavy and light chains were sequenced on the MiSeq Illumina NGS platform using a V2 2X150 Nano flow cell. A read depth predicted to suffice for identifying immunoglobulin genes from hybridomas [48] [between 10,000 and 200,000 reads (**S2 Table**)] was obtained. Using Trinity [57], reads were *de novo* assembled into contigs representing putative transcripts. These transcripts were filtered to detect the presence of a 3'-ISP sequence, then aligned against the IMGT *Mus musculus* database of mouse V, D, and J genes using MiGMAP [56] to ascertain identity as an Ig transcript of interest. In **Fig 2A**, contigs from the 8 NGS-hybridoma libraries contained their respective 3'-ISP sequence, demonstrating method fidelity to identify gene-specific, isotype-specific Igs. The final contigs, shown in **Fig 2B**, contained the signature of productive transcripts in all hybridoma clones. **S3** and **S4 Tables** present the productive transcripts for each of the antibodies characterized in this study.

Multiple homologous transcripts, however, were identified in the contigs generated from *de novo* assembly. To evaluate the fidelity of our bioinformatics approach, we examined if the NGS dataset included all cloned transcript sequences identified using the TOPO cloning/ Sanger sequencing method. Using NRC-13810 as proof-of-concept, our approach identified identical productive Ig transcripts as well as additional unique, productive Ig contigs (e.g., **S2 Fig** and **S3** and **S4 Tables** for NRC-13810). For example, though Igκ-1 of NRC-13810 was identified using both methods, Igκ-2 of NRC-13810_Igκ was only identified using NGS (**Table 2**). **Table 2** summarizes IgV reads using TOPO cloning/Sanger sequencing and NGS/ bioinformatics methods. PhoS1/PstS1$^{NRC-2410}$, SodA$^{NRC-13810}$, and KatG$^{NRC-49680}$ contained multiple Ig heavy and light chains, whereas GroES$^{NRC-2894}$ featured a single pair of Ig heavy and light chains. For the hybridomas with multiple IgVs, validation of the correct combination of heavy and light chains encoding a functional Ab was necessary. IgV results that were subsequently validated are provided in the rightmost panel of **Table 2** to facilitate comparison of the methods for accuracy and efficiency of IgV identification. Most of the IgV sequences were identified by both methodologies. However, deep sequencing with NGS, identified putative Ig transcripts with greater sensitivity of detection than TOPO cloning/Sanger sequencing.

## Functional validation of hybridoma IgV$_H$ and IgV$_L$ sequences

Each putative IgV sequence was cloned and isotype-switched into a common IgG2aκ–Fc vector for use in sequence validation (**Fig 3A**). Thus, the antigen-binding property of the secreted recombinant Ab (rAb) lies in the in-grafted IgV sequences (**Fig 3A**) derived from Mtb Ig transcripts. Since glycosylation alters Ig recognition and binding efficiency to native antigens [58, 59], the IgV$_H$ and IgV$_L$ constructs were co-expressed in 293F cells to mimic B cell glycosylation of immunoglobulins (**Fig 3A**). The IgV were validated by demonstrating (1) the assembly and secretion of the recombinant isotype-switched Ig and (2) the secreted Ig elicits antigen-binding of the target Mtb antigen similar to that of the antibody from the parental hybridoma. Antigen-binding specificity and potency were determined through immunoblot and ELISA assays of purified Mtb full-length antigens (Ag). The native Mtb antigens for GroES and PstS1 were obtained from BEI Resources (GroES, NR-14861; PstS1, NR-14859). The recombinant Mtb antigens, KatG and SodA, were expressed and purified by affinity chromatography (**Figs 3B and S5**).

We first explored the fidelity of the in-grafting process by examining the antigen-binding properties after switching into the same isotype. hyAb$^{NRC-2894}$ shares the same isotype as the

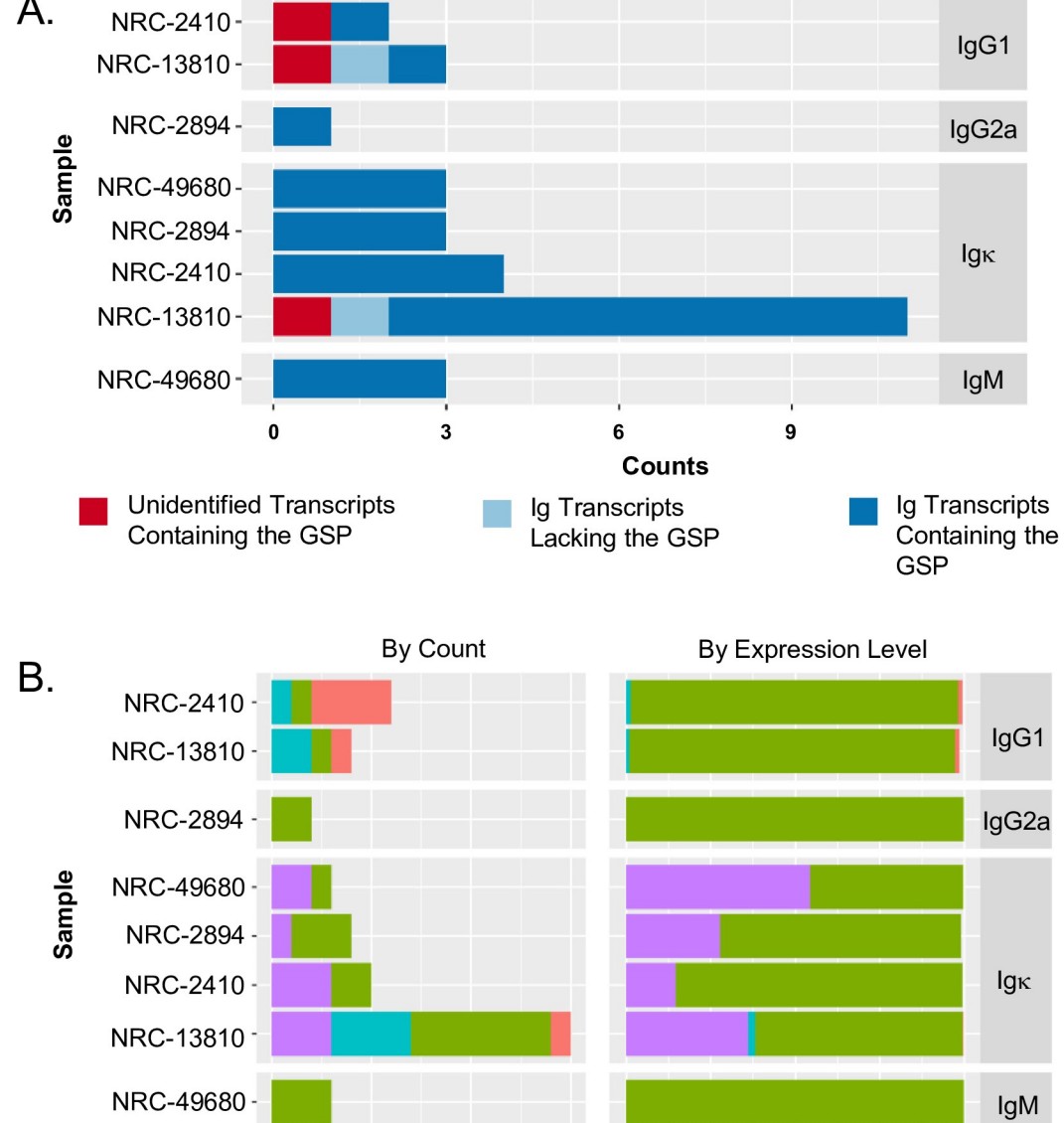

**Fig 2. Hybridoma Ig sequencing using MiSeq Illumina NGS and *de novo* transcript assembly.** Profiles of Contig types among the 4 hybridomas. **A.** Summary of hybridoma sequence contig assemblies identified using isotype-specific ISPs. **B.** The Trinity *de novo* assembled Ig contigs were grouped and quantitated based on counts (Left) or expression levels (Right) into four functional categories: 1) Unique Productive—contigs without STOP codons in the CDRs and found in only one of the tested hybridomas, 2) Unproductive—contigs with one or more STOP codons in the CDRs, 3) Incomplete—only 1 or 2 CDRs identified, and 4) Unidentified—no CDRs identified.

common IgG2aκ–Fc cloning vector, implying antigen-binding prior to and following isotype-switching, if done correctly, should be similar in specificity and affinity. The supernatant of Day-6 post-transfection rAb [NRC-2894] cells was collected. Both recombinant and hybridoma secreted antibodies were highly purified (**Figs 3D and S3B and S4B**). The rAb[NRC-2894]

**Table 2. IgV sequences identified by NGS/bioinformatics and Sanger sequencing methods.**

| Items | Ig Subtypes | transcript tpm* | transcript read count | transcript avg coverage | TOPO-Sanger finds | IgV validated by recombinant ab: antigen-binding assay |
|---|---|---|---|---|---|---|
| SodA[NRC-13810] | heavy, IgG1 | 963725.1311 | 88897.61877 | 10633.68646 | Yes | True |
| | light, Igκ-1 | 39232.47749 | 3635.001071 | 883.7117677 | Yes | True |
| | light, Igκ-2 | 304067.109 | 21428.2634 | 6339.72289 | No | True |
| | light, Igκ-3 | 38357.25488 | 2533.438146 | 783.5375709 | No | False No/little Secretion |
| | light, Igκ-4 | 13688.0569 | 849.329898 | 273.9773865 | No | False No/little Secretion |
| PstS1[NRC-2410] | heavy, IgG1 | 968826.8725 | 40043.93523 | 9073.399221 | No | True |
| | light, Igκ-DIV | 843113.5427 | 20217.92773 | 6151.499309 | Yes | True |
| | light, Igκ-S26 | 3807.603 | 74.34237 | 25.69437 | No | False |
| KatG[NRC-49680] | heavy, IgM | 610640.4988 | 17956.89359 | 4519.352413 | No | True |
| | heavy, IgM_KVS | 378156.7621 | 11071.18364 | 2795.753444 | Yes | False No/little Secretion |
| | light, Igκ | 453166.48 | 9730.178 | 2884.44 | Yes | True |
| GroES[NRC-2894] | heavy, IgG2a | 974745.1045 | 6680.858709 | 1397.669186 | Yes | True |
| | light, Igκ | 47876.02964 | 245.414241 | 74.97380071 | Yes | True |

*: transcripts per million transcripts.

displayed a similar GroES-recognition/binding profile by immunoblot analysis as the parental hybridoma hyAb[NRC-2894] (**Fig 3C**). Immunoblot assay of Mtb whole-cell extracts indicates that both rAb[NRC-2894] and hyAb[NRC-2894] recognize Mtb_GroES as a monomer at ~10 kDa and as a multimeric complex at ~25 kDa (**Fig 3C**). The pair of GroES bands was also recognized by rAb[NRC-2894] with purified GroES proteins (**Fig 3C**). Moreover, both rAb[NRC-2894] and hyAb[NRC-2894] recognize Mtb-GroES over cross-reacting protein in whole-cell extracts from *Yersinia pestis* (**Fig 3C**), indicating specificity for Mtb_GroES. Finally, the GroES-binding potency of hyAb[NRC-2894] is indistinguishable from that of rAb[NRC-2894] when using an ELISA assay with purified GroES antigen (**Fig 3D**). This data validates the IgV sequence identified from both NGS and TOPO cloning/Sanger sequencing strategies for hyAb[NRC-2894], and the functional retention of antigen recognition following Fc engineering.

To pinpoint the correct Ig-pair within each of the PhoS1/PstS1[NRC-2410], SodA[NRC-13810], or KatG[NRC-49680] hybridomas, we followed the same in-grafting process to generate all possible Ig-pair constructs. Each construct pair was separately expressed in 293F cells. Ig culture supernatants were collected at Day-6 post-transfection and examined the presence of secreted Ig using a mouse isotype kit (**Fig 4**). Supernatants with secreted Ig were purified using a series of stepwise acid wash cycles on a protein A column (**S4 Fig**). Spectrophotometric absorbance at A280 identified protein-containing fractions and immunoblotting confirmed the Ig-containing fractions using anti-IgG antibodies (**S4 Fig**). The Ig positive fractions were pooled, and the rAb was concentrated by centrifugation using a spin column concentrator with a 10K molecular weight cutoff. Sufficient purity for comparison against hyAb was confirmed using gel electrophoresis with Coomassie blue or Silver staining (**Fig 4**). To validate rAb, the Ag-Ab binding potential of each $IgV_H/IgV_L$ pair was compared to that of the corresponding hyAb. Equivalent amounts of either the rAb or hyAb were kept in immunoblot and ELISA assays to ensure equitable comparison. The validated rAb 1) recognized purified Mtb antigen in the immunoblot assay and 2) elicited affinity comparable to that of the hyAb in the ELISA assay. Each validated rAb had equal or greater antigen-specific binding activity towards its cognate Mtb antigen as that of the corresponding hyAb (**Fig 4A and 4B**).

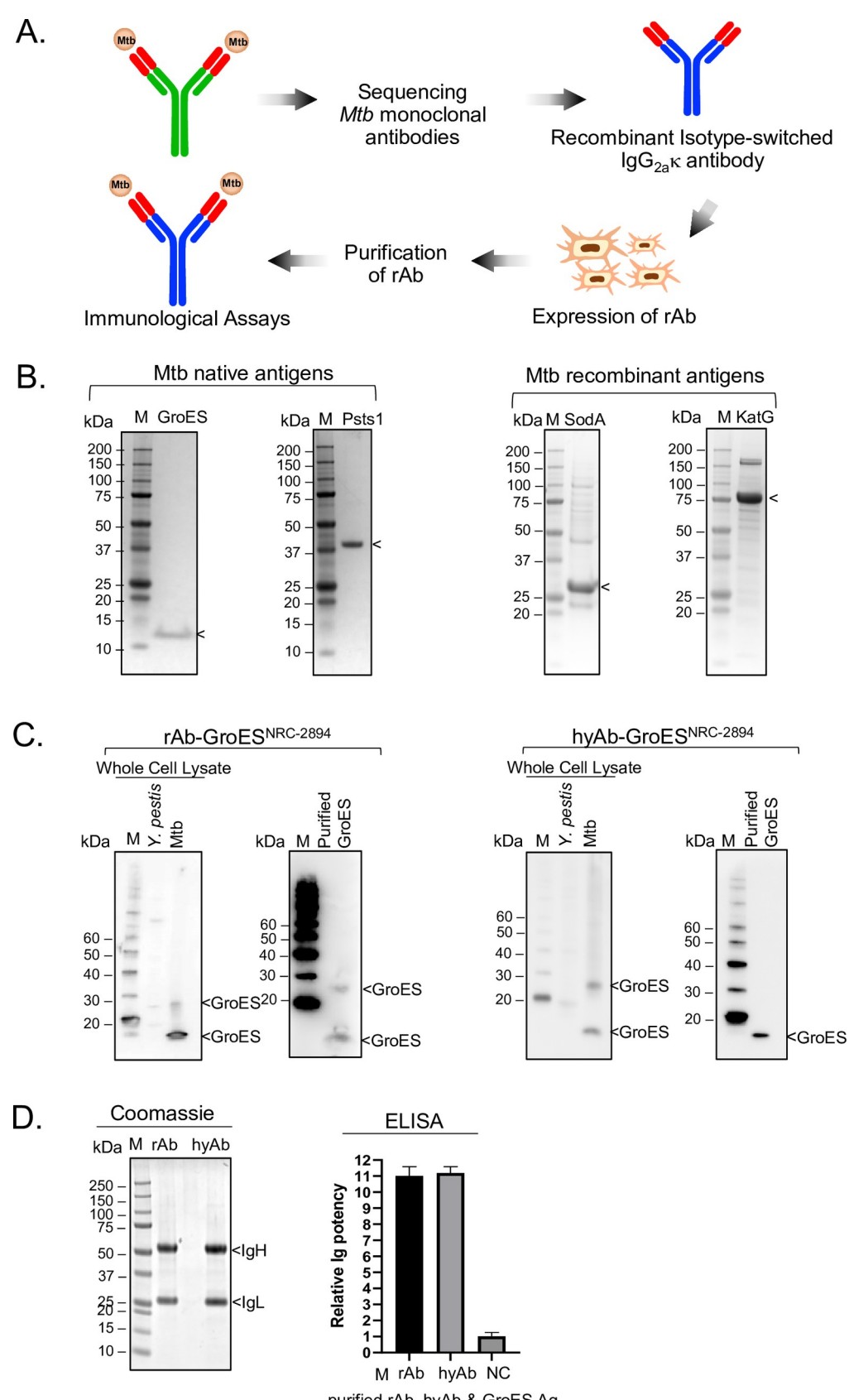

**Fig 3. Confirmation of the fidelity of recombinant validation platform. A.** Diagram depicts the Mtb antibody sequencing and validation workflow. Sequences of the putative IgV (red) were validated by a recombinant Ig platform in which the hybridoma Fc was isotype-switched to a standard mouse IgG2aκ-Fc vector (blue). The recombinant antibody (rAb) was expressed in a mammalian 293F cell line. **B.** Mtb antigens used in the validation study. Coomassie staining of 6 μg of Mtb native antigens (GroES [left panel] or PstS1[right panel]) demonstrated high purity antigens. Recombinant antigens (His-SodA [left panel] and His-KatG [right panel, 2 μg each) were partially purified. **C**. IgV sequence of GroES[NRC-2894] hybridoma was validated. rAb[NRC-2894] elicits a similar GroES-binding profile as its hybridoma parent, hyAb[NRC-2894]. The antigen-binding specificity is assessed by immunoblot against the whole cell extract of Mtb over *Y. pestis*, another human bacterial pathogen. **D**. The antigen-binding property is assessed by ELISA using equal amounts of purified rAb[NRC-2894] or purified hyAb[NRC-2894] to react with the purified GroES antigen. Relative Ig affinity was determined by ELISA endpoint titers normalized against that of an ELISA buffer control (negative control, NC). Data shown here were from two independent experiments performed in triplicate.

While isolation of the highest affinity rAb was achieved for these hybridomas, the data suggests additional complexity in the analysis. Multiple validated rAbs against PstS1 were identified. PstS1-rAb-DIV elicits higher affinity than its hyAb counterpart (**Fig 4A**, Immunoblotting and ELISA), while PstS1-rAb-S26 has affinity equivalent to the negative control. To explain the discrepancies between PstS1-hyAb and PstS1-rAb-DIV, either the Fc engineering enhanced PstS1-rAb-DIV binding activity or the hybridoma secreted a mixture of high- and low-affinity antibodies. Since both rAbs can be secreted but PstS1-rAb-S26 lacks significant activity, the later hypothesis is more plausible. To examine if PstS1-hyAb and PstS1-rAb-DIV share similar epitope-binding potential, a direct competition iELISA assay was employed (**S8 Fig**). Since SodA-hyAb, an IgG1κ isotype, has been purified to homogeneity (**S3C Fig**) and shows no cross-recognition to PstS1 antigens (**S8B Fig**), it was selected as the isotype control to monitor non-specific mechanisms at high concentrations of antibody. As shown in **S8A Fig**, PstS1-hyAb inhibits PstS1-rAb binding in a dose-dependent manner, whereas its isotype control, SodA-hyAb, cannot, indicating that the inhibition is highly selective and supports that PstS1-hyAb and PstS1-rAb-DIV bind to identical or proximal epitopes. Multiple validated rAbs against SodA were also identified. SodA-rAb-Igκ1 and SodA-rAb-Igκ2 demonstrated similar antigen-binding potency, despite featuring different functional Ig chains (**Fig 4B**). In contrast, the rAbs for KatG exhibited properties of monoclonal antibody. KatG-rAb-KVS failed to secrete into the cellular supernatant and was a non-productive aberrant chain (**Fig 4C**). The rAb-IgM was validated, featuring binding activity equivalent to that of KatG-hyAb (**Fig 4C**)**.** Taken together, the PstS1 and SodA rAb data support the recent finding that some hybridoma clones are polyclonal and express additional functional Igs with different variable regions [60].

## Determination of CDRs and FRs from validated IgV sequences

The CDRs and FRs for each of the validated IgV sequences were defined using bioinformatics to facilitate potential downstream applications. We used two common Ig-CDR definition algorithms, KABAT [61, 62] and IMGT numbering schemes [63, 64]. The conventional KABAT Ig CDR search routine is based on antibody sequence and is used widely as the CDR delineation standard [61, 62]. The IMGT approach, currently accepted by the World Health Organization-International Union of Immunological Societies, identifies Ig-CDRs and FRs by integrating the KABAT Ig definition with antibody structural considerations [63–65]. The IMGT-Collier de Perles program provides a 2D view for the position of the amino acids in CDR/FR representation. In either program, the interpretation of CDR/FR assignments should be used with caution. Assigned CDR/FR stretches in one scheme can disagree with those from other schemes [66]. Using the bioinformatics search algorithms Ig-Blast [67] and IMGT/VQuest [68], we defined the Ig_FRs/CDRs, as presented in **S6 Fig** (IMGT), and **Tables 3 and S3** and **S4** (KABAT).

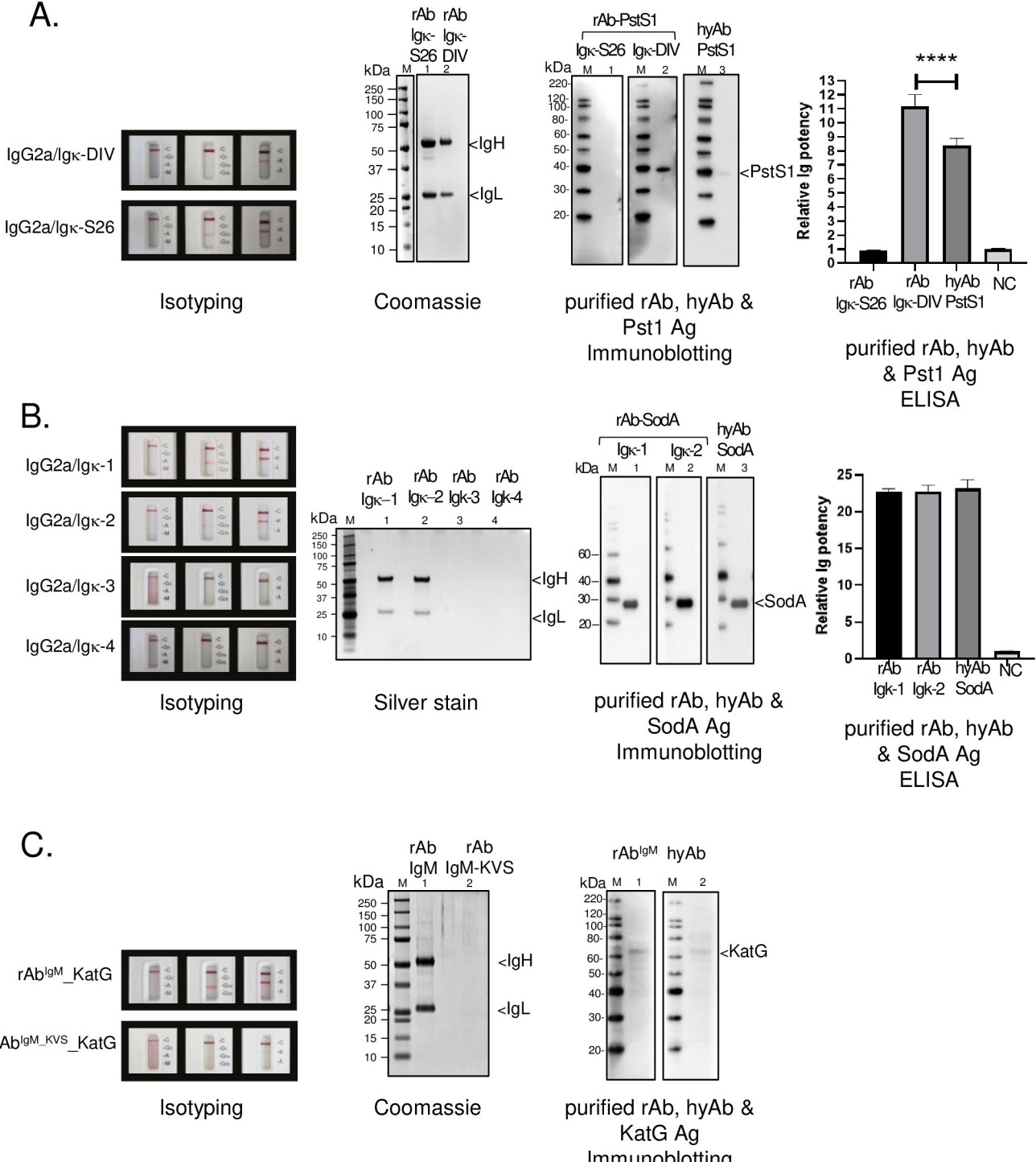

**Fig 4. IgV sequence validation of PstS1^NRC-2410, SodA^NRC-13810, and KatG^NRC-49680 hybridomas.** The correct IgV sequences of PstS1^NRC-2410 (**A**), SodA^NRC-13810 (**B**) and KatG^NRC-49680 (**C**) were identified from multiple IgV_H and IgV_L sequences (**S3** and **S4 Tables**). Each heavy and light chain combination of the isotype-switched IgG2a and Igκ constructs was separately co-expressed in 293 F cells by transient transfection. The presence of an assembled, secreted rAb can be detected in the culture supernatant on Day-6 post-transfection using a mouse Ig Isotype kit (Isotyping label). Some combinations yield an undetectable Ig in the cellular supernatant. The cellular supernatant of each combination was collected and purified by protein A column chromatography. The Ig-equivalent elution fractions were pooled and concentrated. Coomassie or silver staining shows the purity of the rAbs. The Ag-Ab binding potential of each IgV_H/IgV_L pair was compared to that of the corresponding hyAb. Equivalent amounts of either the rAb or hyAb were kept in immunoblot and ELISA assays to ensure equitable comparison. Immunoblot assays detected recognition of their cognate purified antigen

(Immunoblotting). The antigen-binding potency was quantified by ELISA using purified antigens. Relative Ig affinity was determined by ELISA endpoint titers normalized against that of an ELISA buffer control (negative control, NC). Data shown here are from two independent experiments performed in triplicate. P-value<0.001 indicated by the asterisks.

The approach described herein can be applied to the identification and sequencing of functional IgV pair sequences from any hybridoma with certain caveats. First, bioinformatics algorithms and filters should stringently and carefully screen for aberrant or nonfunctional IgV. Second, the potential functional IgV pairs should be validated in their native or recombinant form against the cognate antigen, especially when multiple possible chains are identified. Thus, we have successfully isolated and sequenced the IgV transcripts from Mtb-GroES, Mtb-PstS1, Mtb-SodA, and Mtb-KatG hybridomas. We identified the $IgV_H/IgV_L$ pair that encodes the paratope-determining Ig and defined their CDR/FR regions for each hybridoma.

## Discussion

Tuberculosis (TB) is considered a global infectious disease of urgent concern [4, 5, 69, 70]. Though more than a century has passed since the identification of the bacterial pathogen, no cures for TB exist. Mtb infects around 10 million people annually, resulting in a death rate of 1.4 million, highlighting the need for TB diagnostic and treatment strategies. Ideally, a uniform and robust immune response to infection would lead to a preventive vaccine against TB, but this currently appears unattainable [11, 71–73]. The heterogeneity of host immune responses to Mtb infection manifests in diverse clinical presentations, posing challenges for rapid diagnosis and treatment of TB [74, 75]. The development of antibody-based therapeutic approaches targeted to immunodominant antigens or virulence factors is an important strategy for combating infection.

As an initial step toward engineering candidate antibodies for potential therapeutic and diagnostic applications, the IgV sequences of monoclonal antibodies recognizing Mtb virulence factors were determined. The IgV sequences for cognate Abs from the hybridomas PhoS1/PstS1[NRC-2410], SodA[NRC-13810], KatG[NRC-49680], and GroES[NRC-2894] (**Table 4**) were determined using classical and contemporary sequencing methods, i.e., direct TOPO cloning/Sanger sequencing and next generation sequencing combined with transcript assembly. Since the abundance of transcripts determined through deep sequencing does not strongly correlate with the abundance of Ig protein expression [60], herein, greater than 50X coverage is performed to uncover even less abundant Ig-encoding transcripts. While Sanger sequencing is limited to only a single forward and a single reverse read length (2X coverage), the substantially longer Sanger reads provide a template for orthogonal verification of the transcript assembled from the NGS reads. Therefore, the combination of the two sequencing methods enabled a robust IgV sequencing approach in this study.

As is well-understood, hybridomas may lose expression of functional antibodies through passage [76]. Validation of IgV sequence preserves the functional utility of monoclonal antibodies by capturing the epitope recognition domain and enables generation of recombinant antibodies with equivalent properties. The determination of correct IgV/paratope sequence is a recognized challenge in hybridoma sequencing because of an abundance of aberrant Ig transcripts in hybridomas [49–53, 77]. Unproductive aberrant immunoglobulins (Igs) contain stop codons in the reading frame [51], while productive aberrant Igs achieve full-length transcripts without functional domains [53]. Since IgV sequence identification relies on eliminating these aberrant chains, we created a reference library of aberrant chain sequences for *in silico* subtraction. This library, provided as **S1 Appendix**, includes the overlapping productive aberrant IgVs identified in SP2/0 hybridomas in this work and the earlier literature [49, 50,

**Table 3. Summary of the deduced Ig amino acid sequence of the validated IgV$_H$ and IgV$_L$ sequences.** CDR/FR domains were defined via a KABAT-based algorithm[*].

| Hybridoma | Ig (GenBank#) | FR1 | CDR1 | FR2 | CDR2 | FR3 | CDR3 | FR4 | Antigen |
|---|---|---|---|---|---|---|---|---|---|
| RC-2410 | IgG1 (MW812375) | QVQLQQSGAELMKPGAS VKISCKATGYTFS | GYWVE | WVKQRPGHGLEWIG | EILPGRVSTN YNEKFKA | KATFTADTSSNTAY MQLSSLTSEDSAVYYCAR | FKNYYGSS YNYFDY | WGQGTTLTVSS | PhoS1/ PstS1 |
| | Igκ-DIV (MW812376) | DIVLTQAAPSVPV TPGESLSISC | RSSKSLLHSN GNTYLY | WFLQRPGQSPQLLIY | RMSNLAS | GVPDRFSGSGSGTAF TLRISRVEAEDVGVYYC | MQHLEYPYT | FGGGTKLEIK | |
| NRC-13810 | IgG1 (MW812377) | EVRLEESGGGLVLP GGSMKLSCVASGFTFN | NYWMN | WVRQSPEKGLEWVA | EIRLKSNNYA THYAESVKG | RFTISRDDSKGGVY LQMNNLRAEDTGIYYCTR | EANRGFAY | WGQGTLVTVSA | SodA |
| | Igκ-1 (MW812378) | KIVLTQSPASLAV SLRQRATISC | RASESVDSY GKSFMH | WYQQKSGQPPKLLIY | RASNLES | GVPARFSGSGSRTDF TLTIDPVEADDAATYYC | QQNYEAPRT | FGGGTKLEIK | |
| | Igκ-2 (MW812379) | DIVLTQSPASLAVS LRQRATISC | RASESVDS YGKSFMH | WYQQKSGQPPKLLIY | RASNLES | GVPARFSGSGSRTDFT LTIDPVEADDAATYYC | QQNYEAPRT | FGGGTKLEIK | |
| NRC-49680 | IgM (MW812380) | QVQLKESGPGLVA PSQSLSITCTVSGFSLT | DYGVS | WIRQPPGKGLEWLG | VIWGGGSTYY NSALKS | RLSISKDNSKSQVFLKMN SLQTDDTAMYYCAK | HGNFAY | WGQGTLVTVSA | KatG |
| | Igκ (MW812381) | QIVLTQSPAIMSASLGE RVTMTCTAS | SSVSSSY | LHWYQQKPGSSPKLWIY | STSNLAS | GVPARFSGSGSGTSYS LTISSMEAEDAATYYC | HQYHRSPWT | FGGGTKLEIK | |
| NRC-2894 | IgG2a (MW812373) | EVQLVESGGGLVQPK GSLKLSCAASGFTFK | TYAMN | WVRHTPGKGLEWVA | RIRSKSNNFAT YYADSVKD | RFTISRDDSQSMLYL QMNNLKTEDTAMYYCVK | LTNGYFDS | WGQGTTLTVSS | GroES |
| | Igκ (MW812374) | DIQMTQSPSSLSASLG GKVTITC | KASQDINNYIA | WYQHKPGKGPRLLIH | DTSTLQP | GIPSRFSGSGSGRD YSFSISNLEPEDIATYYC | LQYDNLRT | FGGGTKVEIK | |

[*]S6 Table lists the CDR/FR gene family used for alignment.

53]. After eliminating aberrant IgVs, a minimal and highly refined set of sequence candidates, including all paratope-determining IgVs, were pinpointed. This library may be used whenever performing IgV sequencing in an SP2/0 myeloma background to arrive at, presumably, a similarly small set of sequence candidates for IgV validation.

Multiple IgV heavy or light chain sequences were identified among PhoS1/PstS1[NRC-2410], SodA[NRC-13810], and KatG[NRC-49680] hybridomas (S3 and S4 Tables). IgV sequence validation was needed to identify the functional IgV sequence pairs from each hybridoma. Fc isotype-switched recombinant antibodies were used to validate the paratope-determining IgVs. We showed that the Fc-isotype-switch leaves the IgV paratope unaffected in terms of binding-potency and specificity, as demonstrated by the representative comparison of GroES[NRC-2894] hybridoma recombinant Ig activities (Fig 3C and 3D). The RACE-PCR amplification of this domain, therefore, captured the complete productive and functional IgV sequence. The Ig-pairs within PstS1[NRC-2410], SodA[NRC-13810], and KatG[NRC-49680] hybridomas were also identified using this isotype-switch validation method. The Fc isotype-switched rAbs elicited specific binding to their cognate Mtb antigens with at least the same potency as that of the corresponding hyAb. This approach works whether a single pair of Ig chains, such as in GroES[NRC-2894], is present or when multiple potential IgV$_H$/IgV$_L$ pairs are present. The recombinant Ig expression constructs generated from this study (S5 Table) will be available through NIAID's BEI Resources (www.beiresources.org). The authenticated Mtb-IgV sequences were uploaded to the NCBI nucleotide database (Table 3 and S2 Appendix), meaningfully diversifying the current Mtb IgV sequence collection.

**Table 4. List and features of BEI resources *M. tuberculosis* hybridomas in this study.**

| BEI Resources Item No. | Clone # | Isotype | Antigen | Target gene | PMID |
|---|---|---|---|---|---|
| NRC-13810 | CS-18 | IgG1κ | SodA | Rv03846 | 24586151 |
| NRC-49680 | clone A | IgMκ | KatG | Rv1908c | n/a |
| NRC-2894 | IT-3 (SA-12) | IgG2aκ | GroES | Rv3418c | n/a |
| NRC-2410 | IT-15 (TB72) | IgG1κ | PhoS1/PstS1 | Rv0934 | n/a |

The CDR/FR information provided herein may benefit antibody engineering by guiding the delineation of Mtb Ag-Ab interaction loops. CDRs are considered as the antigen-recognition loops representing the Ab-paratope [45]. Genetic mechanisms, such as VDJ/VJ recombination and somatic hypermutations, generate hypervariability within CDR1-3, and particularly CDR3 [42, 45] separated by conserved framework regions (FR1-4). However, some framework residues can contribute toward antigen reactivity [78, 79]. Hence, the currently assigned CDR/FRs provide a starting point for structural and mutational analyses to determine the key interacting residues in the Ag-Ab binding interface. Knowledge of the minimal but essential Ag-Ab epitope binding residues will enable successful CDR-grafting into humanized antibodies.

The Mtb IgV sequence library and approach described herein will particularly benefit future TB research using epitope binding domains for detection, prevention, and treatment. Antibody engineering can further enhance the epitope-binding affinity of the IgV [80–82]. Increasing evidence points to a protective role for antibodies targeting Mtb virulence-associated factors against TB susceptibility, particularly with IgA in the mucosa [8, 14, 83, 84]. Zimmermann et al. recently demonstrated that an anti-HBHA domain in an IgA isotype is protective, whereas, in an IgG1 isotype background, anti-HBHA exasperates Mtb infection [8]. To further decipher Mtb antibody roles targeting other Mtb virulence-associated factors, antibody engineering that includes Fc-engineering is required. By altering the Fc backbone, changes in Ab effector function are possible, potentially leading to enhanced pharmacokinetics or other biological outcomes such as prolonged Ab half-life in circulation without loss of potency or specificity [80–82]. A simple extension of the robust IgV sequencing and validation workflow described herein may be used to determine and alter the functional IgV sequences of Mtb antibodies targeting Mtb-LAM, Mtb-HBHA, Mtb-Ag85A/B/C complex, and other antigens with therapeutic and diagnostic potential.

## Materials and methods

### Cell cultures and isotyping

The Mtb hybridomas PhoS1/PstS1[NRC-2410], SodA[NRC-1381], KatG[NRC-49680] and GroES[NRC-2894], were obtained from BEI Resources. Hybridomas were generated by chemical fusion of murine SP2/0-Ag14 (SP2/0) myeloma cell lines [85] with the harvested splenocytes from the immunized mice challenged with one of the Mtb antigens PhoS1/PstS1, SodA, KatG, and GroES. Hybridoma clones were grown and expanded in Dulbecco's modified Eagle's medium with 10% fetal bovine serum at 37°C with 5% $CO_2$. Cellular supernatants were collected to determine their antibody isotypes using a mouse antibody isotyping kit (cassette method, Thermo-Fisher Scientific, Waltham, MA; IsoStrip method, Sigma-Aldrich, St. Louis, MO). Cells were grown in Hybridoma-Serum Free Media (SFM) (ThermoFisher Scientific, Waltham, MA) to generate hybridoma antibodies suited for protein G purification.

### RNA extractions

Total RNA from each hybridoma was extracted using a NucleoSpin RNA Plus kit (Takara Inc., Mountain View, CA) following the manufacturer's protocol. RNA concentration and integrity were determined using UV absorbance and RNA gel electrophoresis (Lonza™ Reliant™ Precast RNA Gels, Fisher Scientific Inc.).

### 5' RACE-PCR

mRNA was reverse-transcribed to synthesize first-strand cDNA using a SMARTer RACE 5'/3' Kit (Takara Inc., Mountain View, CA). 5' RACE-PCR was performed according to

manufacturer's protocol (Takara Inc., Mountain View, CA) using 3'-primers specific to highly conserved antibody isotype-specific constant regions [isotype specific primers (ISP); referred to as gene-specific primers (GSP) by the manufacturer (Takara Inc., Mountain View, CA)].

## DNA preparation for TOPO cloning

Variable heavy and light chains of the 4 hybridomas were PCR amplified and isolated using a gel extraction kit (Qiagen Inc., Germantown, MD). IgV RACE-PCR amplicons were cloned into a TOPO sequencing vector (Invitrogen Inc., Carlsbad, CA). Seven to ten transformants were randomly selected for Sanger sequencing by M13 forward and reverse primers (Invitrogen Inc., Carlsbad, CA).

## MiSeq Illumina NGS

Library preparation was performed using ~1 μg of IgV RACE-PCR amplicons and MiSeq® Reagent Micro Kit v2 (Illumina Inc. San Diego, CA). Pairwise-sequencing was performed on an Illumina MiSeq Instrument using a V2 2X150 Nano flow cell (Illumina Inc. San Diego, CA). At least 10,000 reads were obtained per sample (**S2 Table**).

## Bioinformatic analysis of the Ig sequence resulting from direct cloning or NGS-high throughput sequencing

The quality of all input sequence data was confirmed using FastQC, a standard tool for assessing next generation sequencing quality metrics [86]. Reads were then assembled into contigs representing individual transcripts using the RNASeq *de novo* assembler, Trinity v2.6.6 [57], followed by quantification of transcript expression levels using a pseudoalignment approach with Salmon v0.10.0 [87]. To ensure the identification of contigs with a well-defined V(D)J region, we implemented MiGMAP v1.0.3 [56], a wrapper for the NCBI IgBLAST tool v.1.4.0 that adds user convenience functions to the base IgBLAST tool, to align all contigs again the IMGT *Mus musculus* database of mouse V, D, and J genes. Based on these approaches, we grouped these contigs into four categories:

1. Productive contigs—the V(D)J region present without frameshifts or stop codons;

2. Unproductive contigs—the V(D)J region present with one or more frameshifts or stop codons;

3. Incomplete contigs—only a partial V(D)J region present without frameshifts or stop codons;

4. Unidentified contigs—no V(D)J region found.

*De novo* assembled contigs were further searched via an alignment approach using BlastN [88] for the existence of specific ISP primer sequences in the putative constant Ig regions. Finally, a custom R script (**S3 Appendix**) was applied to both the assembled NGS contigs and TOPO cloning/Sanger sequencing datasets to identify the aberrant sequences shared among the hybridoma cell clones examined, including the four hybridomas described in this study and ten additional Mtb hybridomas with an SP2/0 background. After eliminating common aberrant Ig chains, the sequences of the productive contigs were examined and shared little identity with the sequences of well-known aberrant Igs in the literature ([49, 50, 53]. Genbank Accession numbers have been assigned as follows: AF220155, AF220156, AF220157, D14170, D14171, D14173, D50398, S65377, AF089740, AF019945, AF230099, DQ355823, X80944, X80954, AF089742, AF039853, X80944, X80954, AF039853 and M183140.

## Mouse IgG2a Isotype-switched antibody

Synthesis of the mouse IgG2a Isotype-switched IgV chimeras was provided by GenScript Inc. Gene blocks of the putative IgV sequences (FR1/CDR1 to FR4/CDR4, **S3 Table**) were cloned into an intermediate cloning vector, pUC57 and subsequently into GenScript IgG2a/Igκ Fc-expression vectors. Note that the leader sequences of all the recombinant antibodies are MGWSCIILFLVATATGVHS. This leader peptide has been successfully employed in antibody production with 293F suspension cell systems (GenScript Inc. Piscataway, NJ).

## Recombinant antibody production

FreeStyle 293F cells (Invitrogen Inc., Carlsbad, CA) were grown in suspension on a platform shaker (ThermoFisher Scientific, Waltham, MA) in a humidified 37˚C, 8% $CO_2$ incubator with rotation at around 150 rpm. Cultures were maintained for 5–10 passages prior to performing transfections to ensure stable growth patterns. Polyethylenimine (PEI) (25 kDa linear PEI, Polysciences Inc., Warrington, PA) was prepared as a stock solution at a concentration of 1 mg/ml in a buffer containing 25 mM HEPES and 150 mM NaCl (pH 7.5). For best transfection efficiency, cells had a viability of >95% at the time of transfection. Twenty-four hours prior to transfection, cells were split to a density of ~1 x $10^6$ cells/ml and cultured overnight in the $CO_2$ incubator with shaking at 37˚C. Cell density was ~2 x $10^6$/ml at the time of transfection. For transfection, the Ig expression vector plasmid DNAs (1:1 of Ig heavy:light in μg) were added to the cells at a final concentration of 3 μg/ml and PEI at a final concentration of 3 μg/ml of transfection volume (at 1:1, DNA: PEI ratio). After 24 hr, the cells were diluted 1:1 with pre-warmed Freestyle™ 293 Medium supplemented with valproic acid (VPA) (Sigma-Aldrich, St. Louis, MO) to a final concentration of 2.2 mM. Ig enriched cellular supernatants were harvested at day 4–6 post-transfection with viability slightly greater than 55%.

## Recombinant and hybridoma antibody purification

Purification employed a protein A column for recombinant antibodies and a protein G column for hybridoma antibodies (ThermoFisher Scientific, Waltham, MA). Cellular supernatants were added at 1:10 v/v of the binding buffer appropriate for the column (ThermoFisher Scientific, Waltham, MA) to ensure proper ionic and pH conditions for later affinity matrix-binding. The insoluble precipitates were removed by centrifugation at 9,500 rpm for 10 mins at 4˚C. The appropriate column was utilized according to the manufacturer's recommendations for equilibration and sample loading to purify the Ig. The column was washed with 5 column volumes (CVs) of binding buffer prior to stepwise acid washes (2 CVs in each pH buffer, 8.5, 6.5, 5.5) followed by Elution buffer (ThermoFisher Scientific, Waltham, MA) and neutralized using 1/10 (v/v) of 1M Tris/HCl, pH 8.0 (VWR, Radnor, PA). Protein dye staining and gel electrophoresis were employed to identify Ig within collected fractions. The final Ig-yield was determined by BCA protein assay (ThermoFisher Scientific, Waltham, MA)

## Induction of recombinant Mtb antigens expressed in BL21/DE3, pLysS or BL21/DE3

Bacterial expression plasmids encoding the six-Histidine tagged SodA (rSodA) (BEI Resources, Manassas, VA) were transformed into BL21/DE3, pLysS competent cells (Invitrogen Inc., Carlsbad, CA). Bacterial expression plasmids encoding the six-Histidine tagged KatG (rKatG) (BEI Resources, Manassas, VA) were transformed into BL21/DE3 competent cells (Invitrogen Inc., Carlsbad, CA). Once bacterial density reached an A600 of 0.4 to 0.6, 0.2 mM to 1 mM IPTG (ThermoFisher Scientific, Waltham, MA) was added to induce recombinant

protein expression. The culture was harvested after overnight incubation at 37°C with shaking at 200 rpm.

## Recombinant antigen purification

**Recombinant KatG.** After IPTG induction, rKatG-BL21/DE3 bacterial pellets were collected by centrifugation at 9,500 rpm for 10 mins. The rKatG pellets were resuspended in a lysis buffer containing 20 mM Tris, pH 8.0, 500mM NaCl, 30 mM imidazole, and protease inhibitors (Calbiochem, Burlington, MA) at 1:5 ratio (lysis buffer: bacterial culture, v/v) to prepare rKatG lysates. rKatG was extracted by briefly sonicating at an output setting of 10, for 3 to 5 times at 10–20 second intervals in ice to prevent overheating. Insoluble cell debris was then removed by centrifugation at 13,000 rpm for 10 min. The rKatG lysates were incubated with Ni$^{++}$-NTA beads (GE Healthcare, Chicago, IL) overnight at 4°C with gentle mixing to ensure sufficient reaction time. The beads were washed three times with the lysis buffer containing 30 mM imidazole prior to 0.5M imidazole incubation to release rKatG from the beads. The purity of the released rKatG was shown by Coomassie Blue stain on SDS-PAGE gels.

**Recombinant Sod (rSodA).** After IPTG induction, rSodA-BL21/DE3 bacterial pellets were collected by centrifugation at 9,500 rpm for 10 mins. To prepare SodA lysates, the rSodA pellets were resuspended in lysis buffer containing 20 mM Tris, pH 8.0, 500 mM NaCl, 60 mM imidazole, 8 M urea and protease inhibitors (Calbiochem, Burlington, MA) at 1:5 ratio (lysis buffer:bacterial culture, v/v). rSodA was extracted by mixing with lysis buffer at room temperature for 2 hours. Insoluble cell debris was then removed by centrifugation at 13,000 rpm for 10 min. The rSodA lysates were incubated with Ni$^{++}$-NTA beads (GE Healthcare, Chicago, IL) overnight at room temperature with gentle mixing to ensure sufficient reaction time. The beads were washed three times with the lysis buffer containing 60 mM imidazole, 8 M urea prior to 0.5 M imidazole with urea incubation to release rSodA from the beads. The purity of the released rSodA was shown by Coomassie Blue stain on SDS-PAGE gels.

## Immunological assays

**Western blot or immunoblot.** Protein samples were treated with sample buffer containing reducing agents (DTT or β-ME) and boiled at 70°C to 100°C for 10 mins prior to protein gel electrophoresis. The resolving gel was then transferred onto a polyvinylidene fluoride membrane for immunological detection by reacting with specific antibodies. To compare the Ag-Ab binding potential between rAb and hyAb, equivalent amounts of either the rAb or hyAb at 1 μg/ml were incubated with cognate Mtb antigen at 1 μg/lane in an immunoblot overnight at 4°C. Secondary HRP-conjugated antibody [anti-mouse IgG H+L chains (ab6789, Abcam, Waltham, MA) for GroES, PstS1, and SodA; anti-mouse IgM (ab97230, Abcam, Waltham, MA) for KatG] was added, washed and developed using ECL reagents (Cytiva, Marlborough, MA). Positive signals were detected using Imager c600 (Azure Biosystems, Dublin, CA).

**Enzyme-linked immunosorbent assay (ELISA).** An indirect ELISA (iELISA) method was employed to detect the level of antigen-binding of hybridoma and recombinant antibodies. Plates were coated with 2.0 μg/ml of cognate, purified antigens overnight at 4°C, washed with PBS containing 0.1%, v/v, Tween 20 (PBS-Tween), and blocked in 5% non-fat milk (VWR, Radnor, PA) for one hour at room temperature. Then, plates were incubated with serial dilutions of the purified recombinant or hybridoma antibodies, washed in PBS-Tween, and incubated with horseradish peroxidase (HRP)-conjugated anti-mouse IgG antibody (Abcam, Cambridge, United Kingdom) prior to the addition of tetramethylbenzidine substrate (TMB, ThermoFisher Scientific, Waltham, MA). Spectral absorbance of plates at A650 was determined by microplate reader after addition of TMB blueStop (SeraCare Life Science, Gaithersburg, MD).

A directly competitive iELISA method was employed to measure the relative epitope-binding potential between rAb-Pst1 and hyAb-PstS1. Plates were coated with 1.0 μg/ml of PstS1, purified antigen overnight at 4˚C, washed with PBS containing 0.1%, v/v, Tween 20 (PBS-Tween), and blocked in 5% non-fat milk (VWR, Radnor, PA) for one hour at room temperature. The rAb-Pst1, an IgG2a subclass, was kept at a constant concentration of 100 ng/ml, while the concentration of hyAb-PstS1, an IgG1 subclass, was varied from 0 to 1000 ng/ml. The rAb-PstS1 was selectively recognized by the HRP-conjugated anti-IgG2a antibody (A10685, Invitrogen, Carlsbad, CA). Total binding of either IgG1 or IgG2a was measured by the HRP-conjugated anti-IgG heavy and light chain antibody (ab6789, Abcam, Waltham, MA) in the parallel replicate.

## Supporting information

**S1 Fig. Characterization of the isotypes and quality assessment of the extracted RNA from GroES$^{NRC-2894}$, PstS1$^{NRC-2410}$, SodA$^{NRC-13810}$ and KatG$^{NRC-49680}$.** The extracted RNA from the 4 clones yielded high quality RNA with minimal RNase contamination. **A**. Secreted immunoglobulins from the 4 hybridoma clones and their isotypes were confirmed through isotype cassette by examining their culture supernatants prior to RNA extraction. **B.** RNA purity was determined by a NanoDrop™ 2000 Spectrophotometer. The ratio of $A_{260}/A_{280}$ and $A_{260}/A_{230}$ is greater than 1.9 and 2.0 respectively, indicating high RNA purity. **C**. RNA integrity was assessed by the rRNA ratio of 28S:18S larger than 2-fold after 1.25% formaldehyde, RNA gel electrophoresis. **D**. The degree of RNase contamination was measured by the differential amount of 28S present after (+) and before (-) incubation of the extracted RNA at 37˚C for 2 hours. Minimal RNase contamination was observed.
(TIF)

**S2 Fig. Homology alignment to characterize the IgV found from direct cloning—Sanger methodology using a representative of the 4 hybridoma clones, SodA$^{NRC-13810}$. A.** Homology alignment of the NRC-13810-Ig chains, RACE-IgG1, and Igκ of TOPO TA clones: IgV$_H$ (Left) and IgV$_L$ (Right). **B.** Nucleic acid alignment of the sequence reads against each other. Positions #131- and #111-mark initiation codons of clone #1124 and #1267 PCR-IgV amplicons, respectively. If an IgV transcript contains a STOP codon within its FR/CDR domains, it was classified as an unproductive chain and eliminated. CLC Sequence Viewer v8 was used for Ig alignment.
(TIF)

**S3 Fig.** Purified hybridoma antibody from PstS1NRC-2410 (A), GroESNRC-2894 (B), and SodANRC-13810 (C). Cellular supernatants were collected and incubated with protein G beads overnight at 4˚C. The Ig-bound beads were then gently spun down at low speed and packed into a column. The column was washed with 5 column volumes (CVs) of binding buffer prior to stepwise acid washes (2 CVs in each pH buffer) followed by acid elution (E1-6). The presence of Ig heavy and light chains was examined by gel electrophoresis followed by immunoblot against a polyclonal anti-mouse IgG antibody (top gel). The remaining proteins left on the transferred gels were stained by Silver staining to confirm Ig presence (bottom gel).
(TIF)

**S4 Fig.** Purified recombinant immunoglobulin IgG2aκ from NRC-2410 (A), NRC-2894 (B), NRC-13810 (C), and NRC-49680 (D). Cellular supernatants of the recombinant antibodies were collected and incubated with protein A beads overnight at 4˚C with mixing. The Ig-bound beads were then gently spun down at low speed and packed into a column. The column was washed with 5 column volumes (CVs) of binding buffer prior to stepwise acid washes (2 CVs in each pH buffer) followed by acid elution (E1-9 fractions). The presence of Ig heavy and light

chains was examined by A280 and gel electrophoresis followed by immunoblot against a polyclonal anti-mouse IgG antibody (top gel). The remaining proteins left on the transferred gels were stained by Coomassie or Silver staining to confirm Ig presence (bottom gel).
(TIF)

**S5 Fig.** Mtb Recombinant SodA (A) and KatG (B) were purified by affinity chromatography. Bacterial expression plasmids bearing 6X His-tagged SodA or 6X His-tagged KatG were transformed into BL21, a T7/IPTG expression system. The presence of recombinant proteins was examined by gel electrophoresis followed by immunoblot against an anti-His antibody (top gel). The remaining proteins left on the transferred gels were stained by Coomassie staining to confirm recombinant protein presence (bottom gel). The arrow marks the respective recombinant protein. A. 8M urea was employed to extract rSodA-His from insoluble inclusion bodies. B. Brief sonication of rKatG bacteria pellets in the mild lysis buffer was used to extract rKatG-His.
(TIF)

**S6 Fig. Complementarity determining regions (CDRs) and framework regions (FRs) of IgV$_H$ and IgV$_L$ sequences through IMGT analysis.** Collier de Perles displays of validated Ig heavy (Left) and Ig light (Right) chains of GroES$^{NRC-2894}$ (**A**), PstS1$^{NRC-2410}$ (**B**), SodA$^{NRC-13810}$ (**C**), and KatG$^{NRC-49680}$(**D**) using IMGT/V-QUEST (http://www.imgt.org/IMGT_vquest/vquest) for CDR analysis.
(TIF)

**S7 Fig. RACE-PCR amplicon features as a mix of templates.** NRC-13806, an Mtb hybridoma, was run in parallel with the hybridoma clones from this study and was selected as a representative for the Sanger sequence analysis. Chromatogram (A) or summary (B) reports confirmed the presence of a mixture of DNA template composites within a single RACE-PCR amplicon. Sanger sequencing was performed by ATCC's sequencing facility.
(TIF)

**S8 Fig. Competitive iELISA assay of hyAb-PstS1 against rAb-Igk-DIV. A.** PstS1-rAb-lgk-DIV, an IgG2a subclass, was kept at a constant concentration of 100 ng/ml; while concentrations of PstS1-hyAb (solid lines), an IgG1 subclass, or its isotype control, SodA-hyAb (dash lines), were increased from 0 to 1000 ng/ml in a directly competitive iELISA assay for PstS1 antigen binding. **B**. The specificity of antibody reagents was examined using an iELISA assay with increasing concentrations from 0 to 1000 ng/ml of PstS1-hyAb or SodA-hyAb for PstS1 antigen binding. The PstS1-rAb is selectively recognized using the HRP-conjugated secondary isotype-specific antibody, anti-IgG2a (Red). Total binding of either IgG1 or IgG2a was measured by HRP-conjugated anti-IgG heavy and light chain antibody (Blue) in the parallel replicate. Data shown here from two independent experiments in duplicate.
(TIF)

**S1 Table. Summary of the Isotype Specific Primers (ISP) conservation scores in alignments of Ig-heavy and -light chains.**
(TIF)

**S2 Table. Number of reads per NGS library.**
(TIF)

**S3 Table. Summary of all the Ig nucleic acid sequence (CDR/FR domains defined via a KABAT-based algorithm).** Red asterisks mark the validated IgV$_H$ and IgV$_L$ sequences.
(TIF)

**S4 Table. Summary of the deduced Ig amino acid sequence (CDR/FR domains defined via a KABAT-based algorithm).** Red asterisks mark the validated IgV$_H$ and IgV$_L$ sequences.
(TIF)

**S5 Table. The recombinant constructs generated from this study.** Sequences are available in the BEI-product information sheet.
(TIF)

**S6 Table. The Ig-VDJ gene family used to define the CDR/FR domains in Table 3.**
(TIF)

**S1 Appendix. Aberrant chain library used in this study.**
(ZIP)

**S2 Appendix. Validated, FASTA IgV sequence for Mtb hybridomas.**
(ZIP)

**S3 Appendix. Custom R-script to identify the aberrant sequences shared among most hybridoma cell clones in this study.**
(ZIP)

**S1 Raw images.**
(PDF)

# Acknowledgments

We thank the Karen Dobos laboratory (Colorado State University) for development of Mtb hybridomas deposited with BEI Resources. We thank ATCC personnel including John Cordero and Katherine Wigington for laboratory assistance, Steven King for performing NGS, and Shamim Mohammad and Zhidong Xie for technical advice.

# Author Contributions

**Conceptualization:** Timothy T. Stedman.

**Data curation:** Hui-Chen Chang Foreman, Andrew Frank.

**Formal analysis:** Hui-Chen Chang Foreman, Andrew Frank, Timothy T. Stedman.

**Investigation:** Hui-Chen Chang Foreman, Timothy T. Stedman.

**Methodology:** Hui-Chen Chang Foreman.

**Project administration:** Timothy T. Stedman.

**Supervision:** Hui-Chen Chang Foreman, Timothy T. Stedman.

**Validation:** Hui-Chen Chang Foreman, Timothy T. Stedman.

**Writing – original draft:** Hui-Chen Chang Foreman, Andrew Frank.

**Writing – review & editing:** Hui-Chen Chang Foreman, Timothy T. Stedman.

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
