## [Decision Letter · Decision Letter 0]

2 Jun 2021

PONE-D-21-15036

Determination of variable region sequences from hybridoma immunoglobulins that target Mycobacterium tuberculosis virulence factors

PLOS ONE

Dear Dr. Foreman,

Thank you for submitting your manuscript to PLOS ONE. After careful consideration, we feel that it has merit but does not fully meet PLOS ONE’s publication criteria as it currently stands. Therefore, we invite you to submit a revised version of the manuscript that addresses the points raised during the review process.

Please submit your revised manuscript. If you will need significantly more time than this to complete your revisions, please reply to this message or contact the journal office at plosone@plos.org. Please include the following items when submitting your revised manuscript:

We look forward to receiving your revised manuscript.

Kind regards,

Frederick Quinn

Academic Editor

PLOS ONE

Journal Requirements:

PLOS ONE now requires that authors provide the original uncropped and unadjusted images underlying all blot or gel results reported in a submission’s figures or Supporting Information files. This policy and the journal’s other requirements for blot/gel reporting and figure preparation are described in detail at https://journals.plos.org/plosone/s/figures#loc-blot-and-gel-reporting-requirements and https://journals.plos.org/plosone/s/figures#loc-preparing-figures-from-image-files. When you submit your revised manuscript, please ensure that your figures adhere fully to these guidelines and provide the original underlying images for all blot or gel data reported in your submission. See the following link for instructions on providing the original image data: https://journals.plos.org/plosone/s/figures#loc-original-images-for-blots-and-gels.

PLOS requires an ORCID iD for the corresponding author in Editorial Manager on papers submitted after December 6th, 2016. Please ensure that you have an ORCID iD and that it is validated in Editorial Manager. To do this, go to ‘Update my Information’ (in the upper left-hand corner of the main menu), and click on the Fetch/Validate link next to the ORCID field. This will take you to the ORCID site and allow you to create a new iD or authenticate a pre-existing iD in Editorial Manager. Please see the following video for instructions on linking an ORCID iD to your Editorial Manager account: https://www.youtube.com/watch?v=_xcclfuvtxQ

We note that you have included the phrase “data not shown” in your manuscript. Unfortunately, this does not meet our data sharing requirements. PLOS does not permit references to inaccessible data. We require that authors provide all relevant data within the paper, Supporting Information files, or in an acceptable, public repository. Please add a citation to support this phrase or upload the data that corresponds with these findings to a stable repository (such as Figshare or Dryad) and provide and URLs, DOIs, or accession numbers that may be used to access these data. Or, if the data are not a core part of the research being presented in your study, we ask that you remove the phrase that refers to these data.

Reviewers' comments:

Reviewer's Responses to Questions

**Comments to the Author**

1. Is the manuscript technically sound, and do the data support the conclusions?

Reviewer #1: Yes

Reviewer #2: Yes

2. Has the statistical analysis been performed appropriately and rigorously? 

Reviewer #1: Yes

Reviewer #2: N/A

3. Have the authors made all data underlying the findings in their manuscript fully available?

Reviewer #1: Yes

Reviewer #2: Yes

4. Is the manuscript presented in an intelligible fashion and written in standard English?

Reviewer #1: Yes

Reviewer #2: Yes

5. Review Comments to the Author

Reviewer #1: In this paper authors have described their work on the determination of variable region sequences of hybridoma immunoglobulins (hyAbs) targeting four Mtb proteins (termed as “virulence factors”): PstS1, GroES (both provided by BEI), SodA and KatG (both plasmids provided by BEI. Corresponding hybridomas were also provided by BEI). Putative IgV CDR and FR regions of each hybridoma were sequenced by Sanger and NGS methods to identify all potential IgV (H and L) sequences. Sequences encoding aberrant Ig chains were eliminated through bioinformatics. Retention of paratope sequences were confirmed by isotype-switching of the putative IgV in a common Fc backbone (this work was outsourced). Each H and L chain combination of the isotype-switched constructs was co-expressed and each secreted recIgV was validated (using antibody from the parental hybridoma as reference) for binding to the target antigens. Authors conclude to have successfully isolated and sequenced the IgV transcripts from the 4 hybridomas, and also to have identified the CDR and FR regions in the IgV H/L pairs. The work is interesting since, as the authors have stated, extension of this workflow may help determine/alter the IgV sequences of Mtb antibodies targeting other antigens with therapeutic or diagnostic potential. Nonetheless, readers will be benefitted if following concerns could be addressed.

1. It is stated (with the help of Figures 3 and 4) that the epitope binding potency and specificity of IgV paratopes were unaffected by Fc-isotype switch. Even so, the concern remains whether both antibodies (rAb and hyAb) are binding to the same epitope. One way to address this is to allow one of them to compete with the other during immunoblotting and ELISA.

2. As the authors have stated, glycosylation at specific sites plays a very important role in overall function of the antibody. Hence, they have coexpressed the IgVH and IgVL constructs in 293F cells. However, have they also checked whether the glycosylation actually happened in the rAb and whether it was equal or equivalent to that seen in hyAb?

3. The CDRs and FRs for each validated IgV sequence were defined by bioinformatics, using KABAT/IMGT algorithms. However, not all the positions in the traditionally defined CDRs are important for binding. Many positions that contribute critically to the binding energy may reside outside of the CDRs. Moreover, different CDR identification methods may often identify radically different stretches. This and other limitations of the work may be stated in a paragraph under Discussion.

4. It will be helpful if all bioinformatic tools used in this study are briefly described in the ‘Materials and Methods’ section.

5. More details on Immunoblotting will be helpful for the reader.

Reviewer #2: The manuscript by Chang Foreman et al entitled, “Determination of variable region sequences from hybridoma immunoglobulins that target Mycobacterium tuberculosis virulence factors” describes the process to identify the specific variable domains responsible for binding against SodA (Superoxide Dismutase), KatG (Catalase), PhoS1/PstS1 (regulatory factor), and GroES (heat shock protein) from the hybridomas. This is a good study as hybridoma antibodies have been reported to present multiple antibody sequences in some cases. The authors applied RACE-PCR to amplify the antibody domains after identifying the isotypes and was subsequently sequenced. Sequence analysis using both NGS and Sanger sequencing were compared. The report highlights the depth and advantage of NGS method to identify the key domains from a hybridoma for conversion to functional recombinant scFv antibodies. Overall, the study is technically sound with sufficient validation to support the conclusion of the manuscript. I only have some comments to the authors which are listed below.

Comments:

Table 3: The domain sequences would read clearer if the CDR and FR sequences are expressed in amino acid sequence rather than gene sequence. The gene family for the VDJ used should also be detailed.

Line 341: please rephrase ready detection.

As the authors mentioned that deep sequencing capability is preferred to uncover even less abundant Ig-encoding transcripts, was gene analysis or VDJ gene analysis done to identify the clonality of the hybridoma?

Line 450 to 452: Finally, a custom R script was applied to both the assembled NGS contigs and TOPO cloning/Sanger sequencing datasets to identify the aberrant sequences shared among most hybridoma cell clones. Is the script provided?

Line 363, Since IgV sequence identification relies on eliminating these aberrant chains, we created a reference library of aberrant chain sequences for in silico subtraction. How was this library characterized and validated? There is no mention of the details to the reference library in terms of the selection process, characterization, and validation.

There is the added point that the sequence information provided would also help to improve structural studies especially those on antibody-antigen interactions which should be added in the discussion to highlight the importance of the study.

In addition, please provide a percentage of coverage when using Sanger vs NGS. Also discuss the potential reasons/complications that resulted in a lower coverage using Sanger.

The apparent V,D,J segments used and their combination should be mentioned and discussed. Also, are the combinations commonly used in TB.

There are some spelling and grammatical errors which would benefit from another round of editing.

Overall, the paper shows an interesting concept of generating scFv from hybridomas with a specific interest on MTb targets. The message that could be highlighted additionally is the flexibility and coverage provided by NGS as well as the potential identification of multiple sequences from a hybridoma which indicates the presence of multiple clonality.

6. PLOS authors have the option to publish the peer review history of their article (what does this mean?). If published, this will include your full peer review and any attached files.

Reviewer #1: **Yes: **Sudhir Sinha

Reviewer #2: No

---

## [Author Response · Author response to Decision Letter 0]

11 Jul 2021

PONE-D-21-15036

Determination of variable region sequences from hybridoma immunoglobulins that target Mycobacterium tuberculosis virulence factors

PLOS ONE

> We want to express our thanks to both reviewers and to the editor for the encouraging feedback and the opportunity to revise our manuscript. We are grateful for the thorough and careful review. We have addressed the concerns in a point-by-point response below. This thoughtful review process has improved the revised manuscript, and we hope you find it worthy for publication. Please note that all manuscript line number changes addressed below refer to text visible when “No Markup” is selected in the “Track changes” tab of Microsoft Word.

and

> Our manuscript meets PLOS ONE’s style requirements, including those for file naming.

In your cover letter, please note whether your blot/gel image data are in Supporting Information or posted at a public data repository, provide the repository URL if relevant, and provide specific details as to which raw blot/gel images, if any, are not available. Email us at plosone@plos.org if you have any questions

> The original blot/gel image data are now provided in Supporting Information.

> The statement “Recombinant KatG was unstable in this strain as a His-tag antibody detected multiple degradation products (data not shown)” has been removed.

Reply to the Reviewer #1 (Dr. Sinha)

1. It is stated (with the help of Figures 3 and 4) that the epitope binding potency and specificity of IgV paratopes were unaffected by Fc-isotype switch. Even so, the concern remains whether both antibodies (rAb and hyAb) are binding to the same epitope. One way to address this is to allow one of them to compete with the other during immunoblotting and ELISA.

> Although rAb and hyAb exhibit similar affinity and specificity in the antigen recognition, direct analysis of the interface between paratope and epitope has not been performed. Therefore, we changed the wording of “epitope-binding” to “antigen-binding” throughout the text. We agree that Figure 3 and 4 only demonstrate the antigen-binding potency of rAb and hyAb. When biochemical features such as affinity and specificity are similar (Figs 3C-3D and Figs 4B-C), the underlying molecular mechanisms (e.g. paratope:epitope interaction), are often conserved. 

However, to address Dr. Sinha’s question on measuring the epitope-binding potential between rAb and hyAb, we chose PstS1-hyAb and its rAb as an exemplar. Among the 4 hybridoma clones, the PstS1 clone elicited the most notable, differential affinity between rAb and hyAb. We examined if rAb and hyAb of PstS1 can compete for PstS1 in a direct competitive iELISA assay (Fig 8s). With PstS1 antigen as the limiting reagent (Fig 8s, blue), PstS1-hyAb, a lower affinity antibody than rAb, can still outcompete PstS1-rAb in PstS1-binding, eliminating almost all PstS1-rAb binding, clearly indicating that both PstS1-rAb and -hyAb recognize co-localized or identical PstS1 epitopes. This result supports our hypothesis that PstS1 hybridoma is polyclonal, secreting a mixture of antibodies with varied affinities.

We have added the antibody competition result in the Result section from line 302 to line 305 in the edited text, and description of the method in the Materials and Methods section from line #573 to line #579 of the edited text.

2. As the authors have stated, glycosylation at specific sites plays a very important role in overall function of the antibody. Hence, they have coexpressed the IgVH and IgVL constructs in 293F cells. However, have they also checked whether the glycosylation actually happened in the rAb and whether it was equal or equivalent to that seen in hyAb? 

> As the reviewer mentioned, glycosylation is very important for antibody function. The majority of glycan studies center on the glycosylation impact on Fc-mediated antibody effector functions; however, a few studies mention glycan impact on Ab stability and recognition. Though our focus lies in the paratope determining IgV sequences, and not the Fc-region, expression of rAbs in 293F with post-translation modifications mimicking that of mammalian B cells avoids possible issues such as Ab aggregation, secretability, and/or alterations in antigen-binding. As demonstrated in Figs 3c and 3d, rAbGroES expressed in 293F cells featured comparable affinity and specificity toward Mtb-GroES as hyAbGroES. 

We did not compare the glycan profiles between the rAb and hyAb, since the validation of rAbGroES supports that our expression system preserves enough antibody features for interrogating IgV-determining paratope properties. However, we agree that knowledge of the glycan profiles between rAb and hyAb could benefit future Ab production for commercialization. However, glycan profiling will require more than simple sugar staining. Genetic approaches coupled with Mass Spectromery analysis may be necessary and lie outside the scope of this current paper. 

3. The CDRs and FRs for each validated IgV sequence were defined by bioinformatics, using KABAT/IMGT algorithms. However, not all the positions in the traditionally defined CDRs are important for binding. Many positions that contribute critically to the binding energy may reside outside of the CDRs. Moreover, different CDR identification methods may often identify radically different stretches. This and other limitations of the work may be stated in a paragraph under Discussion.

> The reviewer’s points are well-taken. We have added caution statements about interpreting CDR/FR stretches in line #318 to #323 of the Result section under “Determination of CDRs and FRs from validated IgV sequences”. The application of CDR/FR information was added in line #396 to #404 of the Discussion section.

4. It will be helpful if all bioinformatic tools used in this study are briefly described in the ‘Materials and Methods’ section.

> We have provided additional description of the tools and modified the Materials and Methods, lines #455-462, as well as included our custom R-scripts in Appendix 3s. 

5. More details on Immunoblotting will be helpful for the reader.

> We have added immunoblotting details in the Materials and Methods, line #555 to line #561.

 

Reply to the Reviewer #2

1. Table 3: The domain sequences would read clearer if the CDR and FR sequences are expressed in amino acid sequence rather than gene sequence. The gene family for the VDJ used should also be detailed. 

> We have correspondingly changed Table 3 and added Table 6s to annotate the gene family-associated Kabat alignments to define CDR/FR stretches. 

2. The expression in Line 341: please rephrase ready detection.

> We have rephrased the statement to “The heterogeneity of host immune responses to Mtb infection manifests in diverse clinical presentations, posing challenges for rapid diagnosis and treatment of TB.”

3. As the authors mentioned that deep sequencing capability is preferred to uncover even less abundant Ig-encoding transcripts, was gene analysis or VDJ gene analysis done to identify the clonality of the hybridoma?

> Thank you for pointing out that the NGS workflow could be extended to investigate hybridoma clonality. NGS offers high throughput, high fidelity and deep sequencing capability, perfect for sequence annotation of VDJ. Deciphering the molecular mechanisms driving genome heterogeneity, of course, is very interesting and could be beneficial for the study of B-cell derived tumors. However, we focus on deciphering the hybridoma’s IgV-transcript sequences responsible for encoding the functional, antigen-binding antibody. The source of aberrant chains and/or the mechanisms giving rise to aberrant chains or multi-clonality lies outside the scope of the current effort. 

4. Line 450 to 452: Finally, a custom R script was applied to both the assembled NGS contigs and TOPO cloning/Sanger sequencing datasets to identify the aberrant sequences shared among most hybridoma cell clones. Is the script provided? 

> The script is now provided in Appendix 3s.

5. Line 363, Since IgV sequence identification relies on eliminating these aberrant chains, we created a reference library of aberrant chain sequences for in silico subtraction. How was this library characterized and validated? There is no mention of the details to the reference library in terms of the selection process, characterization, and validation.

> The reference library of aberrrant chain sequences was built from a large IgV- Mtb hybridoma sequencing project. All hybridoma clones were made from fusion of SP2/0 myeloma cells. The 4 clones described in this report were members of that project. All hybridoma cell lines were sequenced using both TOPO and NGS/Bioinformatic methods. The shared IgV contigs between hybridoma clones were collected as common aberrant chain sequences, since each hybridoma clone was known to secrete antibodies recognizing distinct Mtb antigens. The known IgV aberrant chains collected from the published literature were used to populate the the aberrant chain reference library, described in lines #476-479. After filtration, none of the reported IgV sequences share sequence identity with the members of our aberrant library. We have added additional details to describe how we generated the aberrant library in lines #473-474.

6. There is the added point that the sequence information provided would also help to improve structural studies especially those on antibody-antigen interactions which should be added in the discussion to highlight the importance of the study.

> Thank you for the suggestions. We have added these points in Discussion lines #396-404. 

7. In addition, please provide a percentage of coverage when using Sanger vs NGS. Also discuss the potential reasons/complications that resulted in a lower coverage using Sanger.

> The definition of coverage, also called depth of sequencing, is the number of reads aligned to a particular base in a target sequence, typically averaged across all bases in a given target sequence. The necessary coverage for NGS to uncover a TRUE IgV is hard to predict, since it depends on the expression level of the target transcript and the variable size of the IgV transcriptome arising from a mixture of true and aberrant chains. In other studies, coverages from 10X to 30X is sufficient to study human mutations or SNPs. Our sequencing depth reached greater than 50X. The paratope-determining IgV in each clone was successfully identified in this report, indicating that 50X coverage is sufficient depth for hybridoma IgV sequencing.

> The sequencing principles of Sanger and NGS are different. NGS employs a parallel sequencing methodology, whereas Sanger sequencing is a clonal sequencing method. Once a sequencing clone is selected, Sanger sequencing is capable of producing one forward read and one reverse read, which depending on read overlap, can only produce between 1-2X coverage (considering potential strand read discrepancies). The advantage of the Sanger method resides in its longer, continuous reads, as noted in lines #166-170, which typically can reach up to 1kb. In our IgV amplicon, averaging around 500-600 bp, the IgV sequence was determined using a forward and a reverse primer. Thus, all IgVs reported here using the Sanger method have 2X coverage. Detection sensitivity of TRUE IgV lies in the initial selection of TOPO clones. Herein, we randomly selected 7-10 clones in each TOPO-IgV cloning. As shown in Table 2, many failed to uncover a true IgV sequence. Clearly, NGS provided a much more robust and greater sensitivity of detection, as previously described in lines #205-207. We have modified/added lines #354-361 in Discussion to emphasize the superiority of the NGS method in uncovering TRUE IgV.

8. The apparent V,D,J segments used and their combination should be mentioned and discussed. Also, are the combinations commonly used in TB

> As described in line #122 and Fig 1A, our PCR approach to isolate an IgV amplicon employed a 5’-universial prime and a gene (isotype) specific primer (ISP). The ISP aligned with the conserved sequence of CH, Constant region of Ig chain (Table 1 & 1s) so that multiple primers aligned with V-D-J gene segments were unnecessary for amplification. We avoid a multiplex PCR method which frequently gives rise to a distorted representation of immune repertoire due to primer bias. No segments or combinations were used, so no discussion is possible.

> Mycobacterial antigens trigger a diverse array of B cells-VDJ recombination. Heavy chains of IgM, IgA, IgE, IgG1, IgG2, IgG3 isotypes/subclasses in combination with either light chains of kappa or lambda can be found in human TB patients. In mice, Mtb-antigens induce similar humoral profiles consisting of IgM, IgA, IgG1, IgG2a, IgG3, kappa and lambda. Nevertheless, we cannot comment on combinations in the manuscript since we did not employ them.

9. There are some spelling and grammatical errors which would benefit from another round of editing.

> Thank you. We have made corrections in Tables (fonts) and in the text throughout using tracked changes in Word. We also further confirm our format to PLOS requirements. 

> Please note we have also rearranged the order of presentation of Figures 4A to 4C in the revised Results section.

> We have made corrections in the following Tables/Figures

Table 3: We have changed the data from nucleic acid to amino acid presentation and added a footnote indicating the source of gene family used for the alignment (Table 6s)

Fig 1A. Correction on IgL-mRNA label: change VH to VL

Fig 4B. Correction on Immunoblotting/Mw Labels: 50- to 40- & alignments

Table 6s: Added Table. 

Fig 1s-B. Reformatted the A260/A280 label

Fig 4s-B. Correction on bottom label: Coomassie to Sliver Stain

Fig 6s. Reformatted the NRC-2894_Ig��label

Fig 8s. Added Figure. 

Appendix 3s: Added Appendix

---

## [Decision Letter · Decision Letter 1]

16 Jul 2021

PONE-D-21-15036R1

Determination of variable region sequences from hybridoma immunoglobulins that target Mycobacterium tuberculosis virulence factors

PLOS ONE

Dear Dr. Chang Foreman,

Thank you for submitting your manuscript to PLOS ONE. After careful consideration, we feel that it has merit but does not fully meet PLOS ONE’s publication criteria as it currently stands. Therefore, we invite you to submit a revised version of the manuscript that addresses the points raised during the review process.

Please submit your revised manuscript. If you will need significantly more time to complete your revisions, please reply to this message or contact the journal office at plosone@plos.org. Please include the following items when submitting your revised manuscript:

We look forward to receiving your revised manuscript.

Kind regards,

Frederick Quinn

Academic Editor

PLOS ONE

Journal Requirements:

Reviewers' comments:

Reviewer's Responses to Questions

**Comments to the Author**

1. If the authors have adequately addressed your comments raised in a previous round of review and you feel that this manuscript is now acceptable for publication, you may indicate that here to bypass the “Comments to the Author” section, enter your conflict of interest statement in the “Confidential to Editor” section, and submit your "Accept" recommendation.

Reviewer #1: (No Response)

Reviewer #2: All comments have been addressed

2. Is the manuscript technically sound, and do the data support the conclusions?

Reviewer #1: Yes

Reviewer #2: Yes

3. Has the statistical analysis been performed appropriately and rigorously? 

Reviewer #1: Yes

Reviewer #2: N/A

4. Have the authors made all data underlying the findings in their manuscript fully available?

Reviewer #1: Yes

Reviewer #2: Yes

5. Is the manuscript presented in an intelligible fashion and written in standard English?

Reviewer #1: Yes

Reviewer #2: Yes

6. Review Comments to the Author

Reviewer #1: I am satisfied with authors’ response to my queries/comments but it would be nice if they could address one lingering concern about the data shown as Fig 8S. A large amount of PstS1 hyAb was needed to inhibit the binding of rAb, which raises concern about the specificity of this competitive binding. Could it have arisen from a non-specific mechanism such as steric hindrance? Is it possible to address this doubt with the use of an ‘isotype control’ (same IgG subclass but with no or irrelevant antibody activity)?

Reviewer #2: (No Response)

7. PLOS authors have the option to publish the peer review history of their article (what does this mean?). If published, this will include your full peer review and any attached files.

Reviewer #1: **Yes: **Sudhir Sinha

Reviewer #2: No

---

## [Author Response · Author response to Decision Letter 1]

23 Jul 2021

> We have reviewed our reference for completeness and accuracy. 

1. In manuscript line #340, we removed the reference #71 and replaced it with references #4 & #5. (Reference #71 was not the correct reference)

2. We also check if our references meet PLOS journal format recommendations (https://journals.plos.org/plosone/s/submission-guidelines). We have made the following changes.

2.1. Ref #4: added a Web link

2.2. Ref #5: corrected author information and conformed to journal citation format

2.3. Ref #18: font size adjustment

2.4. Ref #55: conformed to journal citation format

2.5. Ref #67: conformed to journal citation format

2.6. Ref #68: conformed to journal citation format

2.7. Ref #70: conformed to journal citation format

2.8. Ref #71: removed. It is an incorrect reference.

Review Comments to the Author

Reviewer #1: I am satisfied with authors’ response to my queries/comments but it would be nice if they could address one lingering concern about the data shown as Fig 8S. A large amount of PstS1 hyAb was needed to inhibit the binding of rAb, which raises concern about the specificity of this competitive binding. Could it have arisen from a non-specific mechanism such as steric hindrance? Is it possible to address this doubt with the use of an ‘isotype control’ (same IgG subclass but with no or irrelevant antibody activity)?

> Thank you for the suggestion. As we have mentioned in our 1st Rebuttal letter, we have changed the wording from epitope-binding to antigen-binding throughout the text since we did not yet provide a direct analysis on the interface between paratope and epitope in Figs 3 & 4. We performed a competitive assay with isotype controls to address your concern over non-specific competition. In Fig 8S, the competitive assay, we tested whether rAb and hyAb of the same antigen can compete for their recognition epitopes. PstS1 group is of particular interest since the hyAb seems to be around 80% as potent as its rAb. In the previous Fig 8S, we did a quick diagnosis, and it was very promising. To provide a more stringent and quantitative analysis, we have revised our Fig 8S and made the modifications listed below. 

1. We decreased the amount of PstS1 antigen from 2 to 1 ug/ml in the overnight coating to the ELISA plate. The reduced number of epitopes should create more selective binding competition between rAb and hyAb, avoiding/reducing the amount of non-specific binding or Ab absorption. 

2. All antibody and antigen reagents used herein are purified or purchased in purified form, thereby reducing the likelihood of impurities interfering with the readouts.

3. In addition, we also cross-examined all the antibody reagents used herein for their recognition specificity (Fig 8sB). 

4. We also included an hAb’s isotype control in parallel to control for non-specific mechanisms at high concentrations of antibody. hyAb-SodA, an IgG1� isotype, has been purified to homogeneity and is highly selective for its binding antigen (Fig8sB, blue dash). As shown in Fig 8sA, hyAb-PstS1 inhibits rAb-PstS1 binding in a dose-dependent manner, whereas its isotype control, hyAb-SodA, cannot, indicating that the assay is highly selective and depends on specific antibody recognition of the antigenic sites (epitopes). 

5. Quantitatively, our assay is in good agreement with the relative affinities between rAb and hyAb-PstS1 (Fig 4A, similar, within 20% difference). Shown below is the Ag-Ab binding equation: 

• When Ka1 is relatively close to Ka2 (less than 1 log difference; in our scenario, less than 20% difference), Ag-Ab complexes comprise roughly a 1:1 ratio of Ab1- and Ab2-components when Ab1 and Ab2 have equimolar starting concentrations. We observed that the concentration of hyAb-PstS1 antibody required to achieve 50% inhibition is between 64 and 128 ng/ml, quite comparable to the concentration of rAb-PstS1 in the reaction (100 ng/ml). 

• Only by increasing the relative concentration of one antibody over the other (significantly) can one achieve “complete” inhibition of the other’s binding when both Ka1 and Ka2 are similar. 1 log (herein 1ug/ml) higher concentration of hyAb over that of rAb should begin to strongly favor hyAb binding, consistent with our data. 

> We modified lines #302-309 and #577-585 to reflect this result.

---

## [Decision Letter · Decision Letter 2]

30 Jul 2021

Determination of variable region sequences from hybridoma immunoglobulins that target Mycobacterium tuberculosis virulence factors

PONE-D-21-15036R2

Dear Dr. Foreman,

We’re pleased to inform you that your manuscript has been judged scientifically suitable for publication and will be formally accepted for publication once it meets all outstanding technical requirements.

Kind regards,

Frederick Quinn

Academic Editor

PLOS ONE

Additional Editor Comments (optional):

Reviewers' comments:

Reviewer's Responses to Questions

**Comments to the Author**

1. If the authors have adequately addressed your comments raised in a previous round of review and you feel that this manuscript is now acceptable for publication, you may indicate that here to bypass the “Comments to the Author” section, enter your conflict of interest statement in the “Confidential to Editor” section, and submit your "Accept" recommendation.

Reviewer #1: All comments have been addressed

Reviewer #2: All comments have been addressed

2. Is the manuscript technically sound, and do the data support the conclusions?

Reviewer #1: (No Response)

Reviewer #2: Yes

3. Has the statistical analysis been performed appropriately and rigorously? 

Reviewer #1: (No Response)

Reviewer #2: N/A

4. Have the authors made all data underlying the findings in their manuscript fully available?

Reviewer #1: (No Response)

Reviewer #2: Yes

5. Is the manuscript presented in an intelligible fashion and written in standard English?

Reviewer #1: (No Response)

Reviewer #2: Yes

6. Review Comments to the Author

Reviewer #1: (No Response)

Reviewer #2: (No Response)

7. PLOS authors have the option to publish the peer review history of their article (what does this mean?). If published, this will include your full peer review and any attached files.

Reviewer #1: **Yes: **Sudhir Sinha

Reviewer #2: No

---

## [Editor Report · Acceptance letter]

12 Aug 2021

PONE-D-21-15036R2 

Determination of variable region sequences from hybridoma immunoglobulins that target *Mycobacterium tuberculosis* virulence factors 

Dear Dr. Foreman:

I'm pleased to inform you that your manuscript has been deemed suitable for publication in PLOS ONE. Congratulations! Your manuscript is now with our production department. 

Kind regards, 

on behalf of

Dr. Frederick Quinn 

Academic Editor

PLOS ONE